# Not All Tokens Are What You Need: Selective Deepening for Efficient Model Reasoning.

## Abstract

Large language models typically process information via two dominant paradigms, both of which can be inefficient. The first is a brute-force approach that ingests vast streams of tokens with uniform effort. The second, a selective approach exemplified by Retrieval-Augmented Generation (RAG), often flattens inherently structured data—like codebases or API schemas—into a context-agnostic list of vector chunks. Both methods have critical flaws: the former is computationally prohibitive, while the latter destroys the hierarchical information necessary for complex reasoning.

This paper introduces *Selective Deepening*, a new navigational framework for model reasoning that respects and exploits the native structure of data. Instead of retrieving from a flattened pool of information, our method first creates a *structural abstraction*—a computationally inexpensive, low-fidelity "map" of the data that preserves its hierarchy. The model then intelligently *navigates* this map to identify the most relevant areas to "deepen" into. Only after this targeted navigation does the model dedicate its full computational power to analyzing the high-fidelity details of the selected components.

By replacing structure-agnostic retrieval with structure-aware navigation, Selective Deepening enables models to reason more effectively and efficiently. We demonstrate the broad applicability and benefits of this principle across diverse tasks, including function calling and code generation. Our experiments show that this approach not only drastically reduces computational overhead but also yields significant improvements in task accuracy by mitigating the context-degradation problems inherent in existing paradigms.

## 1 Introduction

Human perception is a masterpiece of efficiency. When we enter a room, our eyes do not perform a raster scan of every photon; instead, they execute rapid saccades, fixating on salient objects while maintaining a low-resolution awareness of the periphery. When searching for a book in a library, we scan titles and covers—abstract, low-fidelity representations—before committing to reading the high-fidelity content within. This hierarchical and adaptive allocation of cognitive resources (Wolfe, 1994; Navon, 1977; Itti & Koch, 2001) is fundamental to biological intelligence, allowing us to navigate and reason within a world of overwhelming sensory data.

In stark contrast, Large Language Models often employ information processing strategies that are fundamentally at odds with this principle of efficiency. These strategies generally fall into two categories. The first is a brute-force approach that compels models to ingest entire contexts with uniform effort, triggering the well-documented "lost in the middle" phenomenon (Liu et al., 2023; Hsieh et al., 2024). The second strategy is selective retrieval, most famously represented by Retrieval-Augmented Generation (RAG) (Lewis et al., 2020; Borgeaud et al., 2022; Wang et al., 2023a; Nakano et al., 2021; Guu et al., 2020; Izacard & Grave, 2021; Huang et al., 2023). While aiming for efficiency, standard retrieval mechanisms perform a *lossy compression of the data's native structure*. They shred a logical hierarchy—be it a codebase, an API specification, or a document's chapters—into a flat list of semantically similar but contextually unaware vector chunks. This process creates a "semantic similarity trap," where retrieved information is locally relevant but structurally incorrect (Sarthi et al., 2024; Han et al., 2024; Wang et al., 2023b).

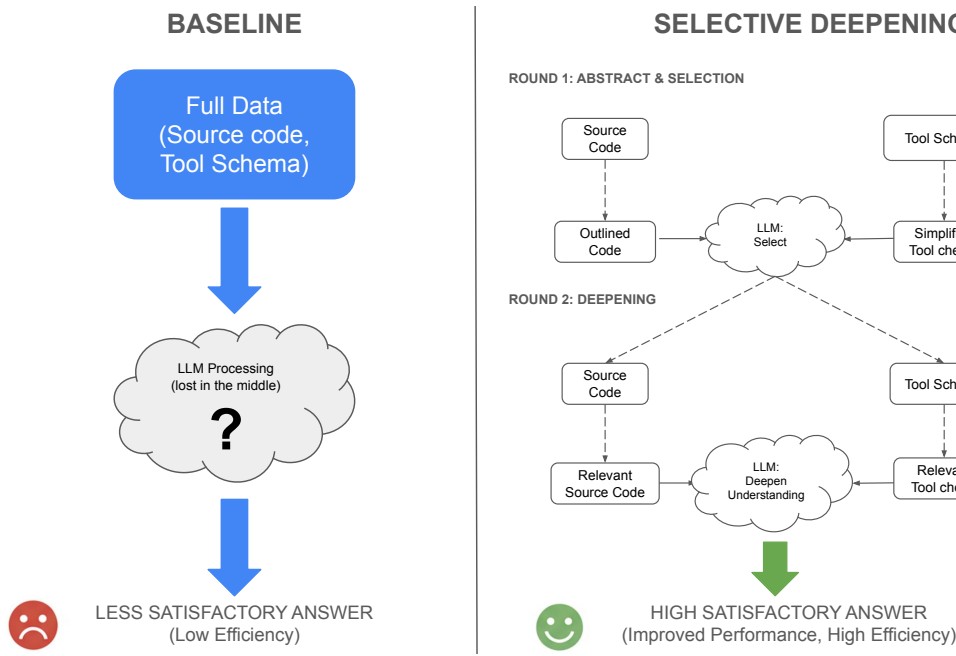

Figure 1: A conceptual comparison between the standard processing paradigm and our proposed **Selective Deepening** approach. **(Left)** In the standard approach, the model's context is filled with all available high-fidelity information (e.g., full tool schemas), leading to high computational costs and the risk of the model getting "lost in the middle." **(Right)** Selective Deepening introduces a two-stage process. First, a fast, low-cost scan of abstract summaries identifies relevant candidates. Second, a focused reasoning stage processes only the high-fidelity details of these candidates, leading to lower costs and higher performance.

We argues that the path to more capable AI requires a shift from simple *retrieval* to structured *navigation*. We introduce the principle of **Selective Deepening**, a framework designed to respect and exploit the inherent hierarchy of information. Instead of pulling from a flat pool of data, Selective Deepening teaches a model to perform a "drill-down" analysis, operating on the inductive bias that relevance is revealed by navigating a data source's explicit structure. As illustrated in Figure 1, it begins with a **structural abstraction**—a low-fidelity map of the information space (e.g., API categories or module outlines). Using this map, the model reasons about the overall architecture to identify the most relevant branches, and only then dedicates its full computational power to the high-fidelity leaves within that selection.

This paper's contributions represent a step towards this new navigational approach:

1. We formalize **Selective Deepening**, a biologically-inspired principle that replaces flat re-
   trieval with structured navigation, enabling models to reason more effectively within com-
   plex information hierarchies.

2. We demonstrate the universality of this principle by instantiating it across diverse and chal-
   lenging domains with inherent structure, including function calling and source code com-
   prehension.

3. We provide compelling empirical evidence that this approach not only yields dramatic gains
   in efficiency but, more importantly, leads to significant improvements in reasoning accuracy
   and robustness by avoiding the pitfalls of structural flattening.

By teaching models to first navigate a structural map and then focus its analysis, we can build sys-
tems that are not only more scalable but also more aligned with the elegant principles of intelligence
found in the natural world.

## 2 THE PRINCIPLE OF SELECTIVE DEEPENING

Selective Deepening is a computational framework that replaces the standard, single-pass processing paradigm with a structured, multi-stage approach. Its design is guided by the core idea of intelligent navigation over inherent data hierarchies, rather than flat retrieval.

### 2.1 A FORMAL NAVIGATIONAL FRAMEWORK

Let us define a general reasoning task where a model $M$ must produce an output $O$ from a query $Q$ and a body of information that possesses an inherent structure. We represent this information source not as a flat set, but as a structured object, $\mathcal{G}$, which can be a tree, a graph, or a timeline. The nodes in $\mathcal{G}$ contain the high-fidelity information chunks $\{i_1, i_2, \ldots, i_n\}$, and its edges represent their structural relationships (e.g., containment, dependency, or temporal succession).

Selective Deepening decomposes the task into a recursive process. The primary instantiation, used in our experiments, is a two-stage application:

**Stage 1: Structural Abstraction & Navigation.** First, the framework creates a low-fidelity *structural abstraction*, $\mathcal{G}' = \text{Abstract}(\mathcal{G})$. This abstraction preserves the essential structure of the original data (the "map") while replacing the high-fidelity node content with computationally inexpensive summaries. This map is then presented to a navigator function, $M_{\text{navigate}}$, which reasons over the query and the data's structure to identify a subset of relevant nodes, $\mathcal{V}_{\text{selected}}$.

$$\mathcal{V}_{\text{selected}} = M_{\text{navigate}}(Q, \mathcal{G}') \tag{1}$$

The resulting candidate set $\mathcal{C}$ is composed of the high-fidelity information from these selected nodes, where $\mathcal{C} = \{i_k \mid v_k \in \mathcal{V}_{\text{selected}}\}$. By design, $|\mathcal{C}| \ll n$.

**Stage 2: Focused Reasoning.** A reasoning function, $M_{\text{reason}}$, subsequently performs the main task, but on a dramatically reduced and structurally-aware context containing the query $Q$ and only the selected high-fidelity candidates $\mathcal{C}$.

$$O = M_{\text{reason}}(Q, \mathcal{C}) \tag{2}$$

The functions $M_{\text{navigate}}$ and $M_{\text{reason}}$ can be implemented by the same underlying model $M$ operating in different passes or by distinct, specialized models.

**Recursive Deepening.** The two-stage process is the base case of an inherently recursive framework. For deeply hierarchical data, the output of $M_{\text{reason}}$ can be a decision to re-apply the principle, treating a selected candidate $c \in \mathcal{C}$ as the new information source $\mathcal{G}_{k+1}$ for a subsequent navigation step. For example, after navigating to a specific class in a codebase, the model could treat that class's method signatures as a new structural abstraction to decide which specific function body to "deepen" into next. The formal exploration of this multi-stage navigation is a compelling direction for future work.

### 2.2 INSTANTIATIONS ACROSS MODALITIES

The power of this framework lies in its application to the intrinsic structure of different data types. The structural abstraction $\mathcal{G}'$ provides the "information scent" needed to guide the subsequent high-fidelity analysis.

**The Nature of Structural Abstractions ($\mathcal{G}'$).** The goal of the abstraction is to provide a computationally cheap but structurally-sound map of the information landscape.

- **For Code Comprehension:** The abstraction is a *symbol tree*, representing the hierarchy of files, classes, and functions with their signatures, stripping away implementation logic.
- **For Structured Data (Function Calls):** The abstraction is a *tool-set schema*, outlining the available tools and their primary purpose, omitting the verbose details of their nested parameter structures.

**The Nature of High-Fidelity Details ($\mathcal{C}$).** Following navigation, $\mathcal{C}$ provides the rich, detailed information from the selected nodes needed for precise reasoning.

- **For Code Comprehension:** The complete source code of the specific functions or files chosen for detailed analysis.
- **For Structured Data:** The full, verbose JSON Schema for the single tool (or small set of tools) selected as a candidate.

## 2.3 Implementation of the Navigator

The practical implementation of the navigator function, $M_{\text{navigate}}$, is a crucial component of the framework. For the experiments in this paper, we focus on a training-free, LLM-based navigator. In this configuration, the Stage 1 prompt contains the user query $Q$ and the serialized structural abstraction $\mathcal{G}'$. The model is then instructed to reason over this map and return the names of the relevant nodes for the high-fidelity analysis in Stage 2. This approach directly leverages the powerful reasoning capabilities of modern foundation models to perform the navigation task. While this training-free implementation offers maximum flexibility, an alternative for specialized domains could involve explicitly fine-tuning the $M_{\text{navigate}}$ to maximize recall on the selection task, ensuring no critical information is overlooked.

## 2.4 Scalability to Massive Corpora: A Hybrid Approach

While the LLM-based navigator is powerful, its performance may be challenging when the structural abstraction $\mathcal{G}'$ itself becomes too large to fit within a model's context window (e.g., a codebase with millions of files). For these massive-scale scenarios, the Selective Deepening framework can be extended with a hybrid navigation strategy.

This approach introduces a preliminary, coarse-grained filtering step. A fast, embedding-based retrieval method can be used to first select a plausible subset of nodes from the enormous graph $\mathcal{G}'$. Crucially, unlike standard RAG which retrieves from a flat list of disconnected text chunks, this step retrieves and preserves the original nodes and their relationships from the structured abstraction. The LLM-based $M_{\text{navigate}}$ then performs its nuanced, reasoning-based navigation on this much smaller, pre-filtered subgraph. This hybrid model combines the raw speed of vector search for initial candidate generation with the superior structural reasoning of our method for the final, precise selection, offering a practical path to scaling Selective Deepening to virtually any size. We consider the empirical validation of this hybrid approach a promising avenue for future research.

## 3 Experiments and Results

Our experiments are designed to validate two core hypotheses: (1) that the Selective Deepening principle can be effectively applied across diverse modalities, and (2) that it yields simultaneous improvements in both task performance and computational efficiency. The primary baseline, which we term **Brute-Force**, provides the model with all available high-fidelity information in a single context window, representing the standard paradigm.

### 3.1 Application to Structured Data: Function Calling

**Setup.** To evaluate Selective Deepening on a structured data reasoning task, we use the Berkeley Function Calling Leaderboard (`BFCL v3`) dataset (Patil et al., 2024). We focus our analysis on the *multiple* and *live_multiple* subsets—with the latter featuring more complex tools from real-world, actively-maintained APIs—as they are specifically designed to test a model's ability to select the correct tool(s) from a list within a single turn. This setup directly probes the challenge of identifying relevant information in a cluttered context. Our experiments were conducted using the Gemma 3 series of models (1B, 4B, and 12B) to assess performance across different model scales.

**Methodology.** The standard approach, which we term the Brute-Force (Baseline), presents the model with the full, verbose JSON schema for all available tools in a single prompt. This includes tool names, descriptions, and detailed parameter specifications.

Our Selective Deepening (SD) method implements the two-stage process defined in Section 2.

1. **Stage 1 (Candidate Selection):** The model is first given the user's query along with a low-fidelity, abstract representation of the tools. We test two levels of abstraction: *Name* only (tool names) and *Name + Description* (tool names with their corresponding descriptions). The model's task is to identify and return a list of relevant tool names.

2. **Stage 2 (Focused Reasoning):** In the second stage, the model receives the user's query again, but this time with the full, high-fidelity JSON schemas for *only* the candidate tools selected in Stage 1. This allows the model to perform detailed reasoning and parameter filling on a clean, focused context.

We measure performance by the function calling Success Rate (SR) and efficiency by the average number of tokens processed per interaction.

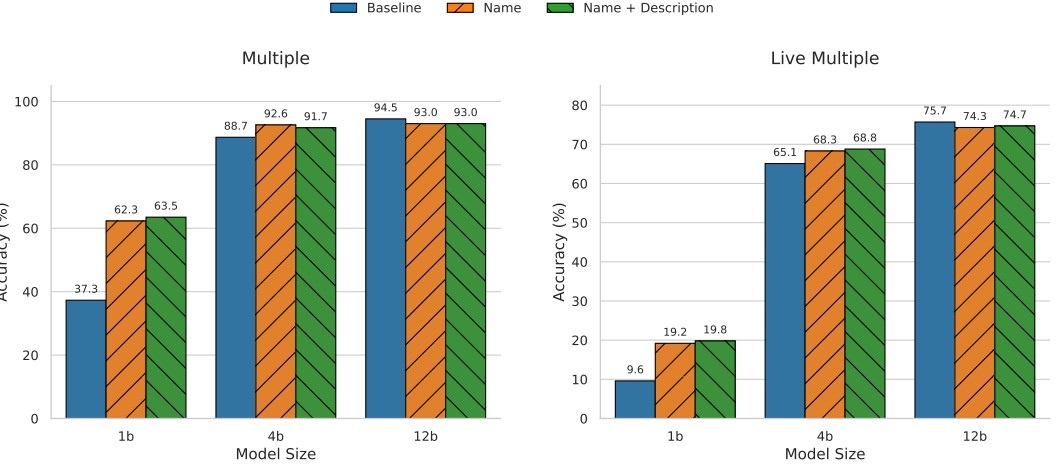

Figure 2: Function calling accuracy on the BFCL v3 *multiple* and *live_multiple* subsets. Selective Deepening strategies (*Name* and *Name + Description*) significantly outperform the *Baseline* across all model sizes. The performance gap is most pronounced for smaller models, which benefit most from the reduced context noise.

**Results and Analysis.**    As illustrated in Figure 2, Selective Deepening yields substantial improvements in function calling accuracy. The key findings are:

- **Superior Accuracy with Less Information:** Both SD strategies consistently outperform the Brute-Force baseline. The improvement is most dramatic for the 1B model, where the baseline struggles with the noisy, verbose context (37.3% SR on *multiple*, 9.6% SR on *live_multiple*). By simplifying the context, our method boosts accuracy to 62.3% on *multiple* and 19.2% SR on *live_multiple*, demonstrating that reducing cognitive load is critical for smaller models.

- **A Revealing Exception in Large Models:** While SD is highly effective, the 12B model shows a slight accuracy decrease on the *multiple* task compared to the baseline. We identified examples where tools could only be differentiated by subtle details in their parameter descriptions, which were absent in our low-fidelity summaries. The 12B baseline, with its larger capacity, was able to leverage this detailed information, while our method filtered it out. This highlights a key trade-off: the efficiency of SD relies on the assumption that the abstract representation retains sufficient information for selection.

Overall, our method achieves superior or highly competitive accuracy on a standard benchmark while inherently improving efficiency by processing only the schemas for selected tools. The full impact of this efficiency gain is analyzed in our robustness tests in Section 3.3.

## 3.2 APPLICATION TO CODE COMPREHENSION

**Setup.** To validate our approach, we evaluated it on a repository-level bug-fixing task using the SWE-bench Lite benchmark (Jimenez et al., 2024). This benchmark contains 300 test instances from popular open-source Python repositories such as Django, Matplotlib, and NumPy. Our experiments were conducted using the DeepSeek-V3[1] API, which has a 64K context length. A core challenge in this task is sifting through a large context of retrieved files to find the precise location for a fix. The SWE-bench framework provides two retrieval settings to benchmark this ability:

- **BM25 Setting:** A practical scenario using the keyword-based BM25 algorithm to retrieve files. SWE-bench Lite provides contexts of varying lengths (13K, 27K, 50K, 100K tokens), which correspond to progressively higher recall rates for the files required for the gold-standard fix (33.7%, 49.0%, 64.7%, and 73.3% respectively). This creates a natural trade-off between having more necessary information and more distracting noise.

- **Oracle Setting:** An idealized scenario where the model is given the exact files modified in the human-authored solution. This isolates the code reasoning challenge by establishing a near-theoretical upper bound on *retrieval* performance.

**Methodology.** We compare our Selective Deepening (SD) method against a standard brute-force **Baseline**, which feeds the entire concatenated content of all retrieved files to the model at once. Our SD method, following the principle in Section 2, employs a two-stage process:

1. **Stage 1 (Outline and Select):** The model first receives a low-fidelity "structural map" of the code, containing file structures and function/class signatures with their implementations hidden. It analyzes this outline to select a small subset of relevant function signatures.

2. **Stage 2 (Focus and Resolve):** The context is rebuilt, revealing the full code bodies for only the selected functions. This focused context is then used to generate the final patch.

Performance is measured by the Resolve Rate (RR), with efficiency measured by API cost.

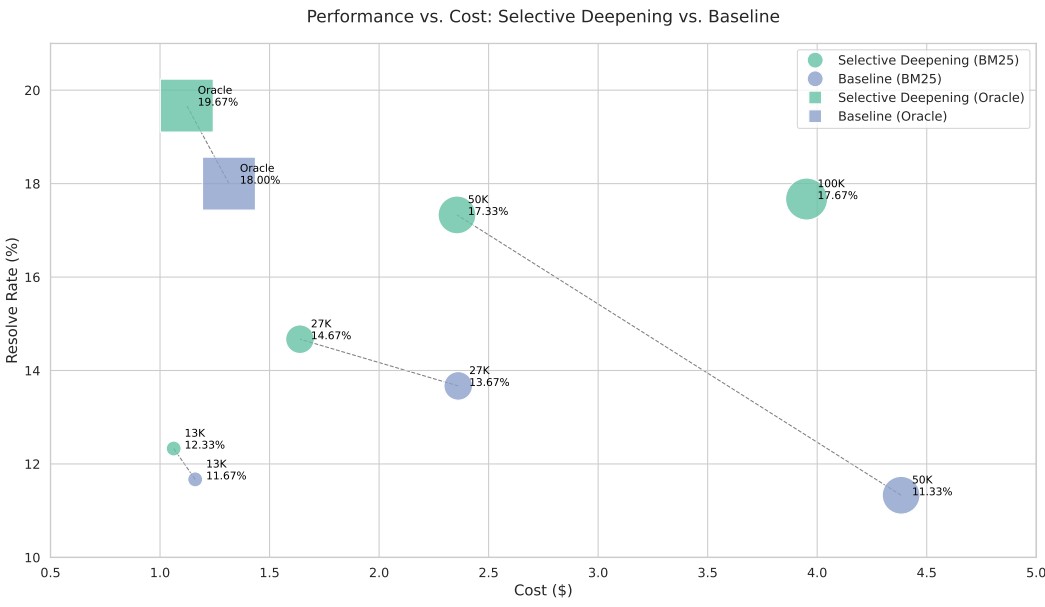

Figure 3: Resolve Rate on SWE-bench Lite as a function of retrieval context size. Selective Deepening's performance scales with increasing context and file recall, while the baseline method's performance collapses under the noise of larger contexts.

[1]We used the official API for the 0324 version.

**Results and Analysis.** Our results demonstrate a clear advantage for Selective Deepening, which consistently outperforms the baseline methods across all settings and context sizes while using significantly less computational cost. As summarized in Figure 3, the key to this success lies in how each method handles the trade-off between useful information and distracting noise as context grows.

- **Translating Recall into Performance:** The central finding is how each method handles the trade-off between information and noise. As context length and recall increase from 13K to 100K, the performance of **Selective Deepening consistently improves**, from 12.33% to 17.67%. This demonstrates it successfully leverages the more complete context provided by higher recall to find better solutions.

- **Baseline Collapse Under Noise:** In stark contrast, the **Baseline** method's performance peaks at the 27K context (14.00%) and then collapses to 11.00% at the 50K context mark. Although the 50K context has a higher recall (64.7% vs. 49.0%), the baseline is overwhelmed by the accompanying noise and its performance regresses. This clearly illustrates the "lost in the middle" (Liu et al., 2024) problem that our method is designed to solve.

- **Superior Scalability and Effective Use of Improved Retrieval:** The baseline's collapse at 50K highlights its practical limitations in noisy settings. Selective Deepening, however, remains robust and effective as context grows. Crucially, its 17.67% resolve rate in the 100K BM25 setting begins to approach the 19.67% achieved in the Oracle setting. This is significant because it shows that as practical retrieval systems improve and provide higher recall, our method is uniquely capable of translating those retrieval gains into tangible improvements in code generation, effectively closing the performance gap caused by imperfect retrieval.

The performance gains are achieved with significant improvements in efficiency. In the 50K dataset scenario, while the baseline must pay the cost to process 50,000 tokens of dense code, our model only processes the high-fidelity code for the few functions it deems relevant. This resulted in our approach achieving its superior accuracy at approximately **half the computational cost** of the baseline, making it a more scalable and practical solution for real-world software engineering tasks.

### 3.3 Stress Test: Robustness, Efficiency, and the Impact of Abstraction

**Setup.** To conclusively validate that Selective Deepening mitigates the "lost in the middle" (Liu et al., 2024) problem, we designed a stress test to analyze model performance under extreme noise. We created a synthetic dataset by augmenting the original toolsets from the BFCL benchmark (Patil et al., 2024) with a varying number of irrelevant "distractor" tools, ranging from 10 to over 200[2]. This experimental design is not merely academic; it mirrors real-world function-calling scenarios where a model must select from a large toolset provided by various MCP services. For instance, a widely used GitHub MCP service exposes 96 distinct tools. This setup, therefore, allows us to precisely measure how each method withstands realistic levels of context pollution.

**Results and Analysis.** The results in Figure 4 confirm the central hypothesis of our work: insulating the reasoning process from noise is critical for success. The performance of the Baseline approach degrades catastrophically as the number of irrelevant tools ($N$) increases. Even the 12B model fails completely when all tools are provided, perfectly demonstrating the "lost in the middle" phenomenon. In stark contrast, Selective Deepening proves exceptionally resilient. Its two-stage design effectively filters the noise, allowing the 12B model to maintain over 50% accuracy in the most cluttered setting—a scenario where the baseline scores 0%.

This experiment also reveals a crucial insight into designing the low-fidelity summaries. By comparing our two SD variants, we find a clear trade-off between a summary's conciseness and its informativeness. In moderately noisy environments ($N \leq 100$), the richer 'Name + Description' summary provides a distinct advantage. However, in extremely noisy environments ($N \geq 200$), these descriptions begin to contribute to the noise, and the ultra-concise 'Name Only' representation

---

[2]We ensured distractor tools had different functionality from the original candidates to maintain correct ground truth.

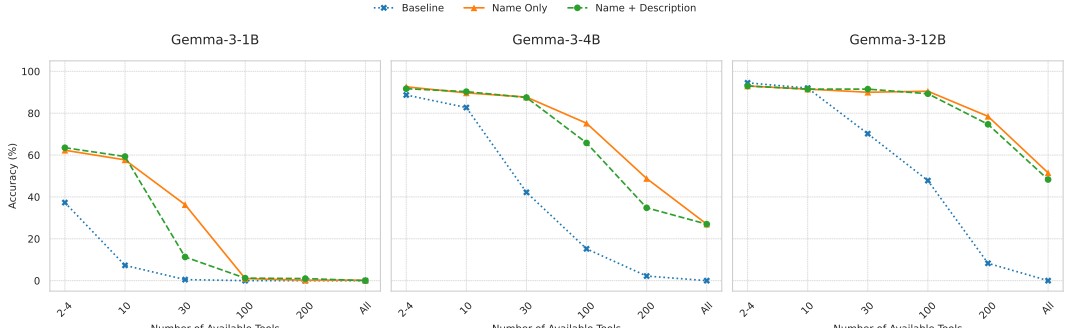

Figure 4: Model accuracy as a function of the number of irrelevant tools in the context. Results are shown for three model sizes. The *Baseline*'s performance collapses as the context is filled with distractors. In contrast, the Selective Deepening methods (*Name Only* and *Name + Description*) demonstrate significantly greater robustness to this noise, with larger models retaining substantial capability even in highly cluttered environments.

becomes more robust. This demonstrates that the ideal abstraction depends on the task's expected signal-to-noise ratio.

This trade-off directly informs the profound efficiency implications of the framework—the ultimate advantage of which is unlocking reasoning over contexts that would be too large for a model's native window or effective context length (Hsieh et al., 2024). The efficiency gains are dramatic. The baseline's cost scales linearly with the number of tools ($Cost_{baseline} \approx N \times Tokens_{full\_schema}$), while our method's cost is an order of magnitude smaller ($Cost_{SD} \approx (N \times Tokens_{summary}) + Cost_{reasoning}$). For example, representing 200 tools would require 30K tokens for the baseline. In contrast, our method requires only 3.6K tokens, with the majority of this cost coming from JSON formatting overhead rather than the summary content itself. Despite this, the approach remains nearly 10x more efficient, making it a highly scalable solution.

## 4 RELATED WORK

The principle of Selective Deepening intersects with several active areas of research. We position our contribution by relating it to three main themes: methods for efficient long-context processing, strategies for selective context reduction, and the historical precedent of hierarchical processing in computer vision.

### 4.1 EFFICIENT LONG-CONTEXT PROCESSING

A primary challenge in scaling language models is enabling them to process ever-longer sequences of tokens. A significant body of work focuses on making the underlying Transformer architecture more efficient. These efforts include engineering innovations in the attention mechanism like FlashAttention (Dao et al., 2022; Dao, 2023) and various sparse attention patterns (Child et al., 2019; Jaszczur et al., 2021; Chen et al., 2024; Ding et al., 2023); algorithmic improvements to positional embeddings that improve length extrapolation (Press et al., 2022; Sun et al., 2023; Su et al., 2023; Chen et al., 2023; Peng et al., 2024); and entirely new, sub-quadratic architectures like Mamba (Gu & Dao, 2023) and RWKV (Peng et al., 2023).

Selective Deepening is orthogonal and complementary to this entire line of research. While these methods optimize how a model efficiently processes a long, flat sequence, our method operates at the data-structuring level. We change *what* information is presented to the model, rather than altering the model's architecture itself. Our principle can therefore be composed with these efficient architectures to yield compounded gains.

## 4.2 SELECTIVE CONTEXT REDUCTION: RETRIEVAL VS. NAVIGATION

A primary approach to long-context challenges is to selectively reduce the information presented to the model, mitigating the "lost in the middle" phenomenon (Liu et al., 2024). The dominant paradigm for this is *Retrieval-Augmented Generation (RAG)*, which has a long history of improving language model performance. RAG has been shown to improve perplexity (Borgeaud et al., 2022; Wang et al., 2023a), factual accuracy (Nakano et al., 2021), downstream task accuracy (Guu et al., 2020; Izacard & Grave, 2021; Lewis et al., 2020), and in-context learning (Huang et al., 2023). Typically combining a standalone retriever (Karpukhin et al., 2020; Wang et al., 2022) with a generator, RAG can be integrated at inference (Khandelwal et al., 2019), fine-tuning (Izacard et al., 2022), or pre-training (Borgeaud et al., 2022), with some end-to-end solutions also proposed (Jiang et al., 2022; Shi et al., 2023).

However, the effectiveness of standard RAG is predicated on **retrieval from a flattened representation**. It chunks a corpus, discarding native structure, which risks retrieving disconnected though semantically similar snippets—the "semantic similarity trap" (Gao et al., 2024)—especially when applied to inherently structured data.

Selective Deepening offers a fundamentally different philosophy: **navigation of a native hierarchy**. Instead of using an external retriever, our method empowers the LLM itself to act as an intelligent navigator. It reasons over a low-fidelity "map" of the data's structure to actively *select* where to focus its attention. This concept of using the LLM for decision-making is related to the field of LLM Agents, where systems like ReAct (Yao et al., 2023) learn to select tools. Selective Deepening can be seen as a specialized form of agentic reasoning where the "action" is the decision to deepen into a specific branch of a data structure. Furthermore, because the structural abstractions are often intrinsic to the data format and the navigation can be guided by a training-free LLM, our method can frequently be implemented via simple prompt engineering, bypassing the complex data pipelines required by traditional RAG systems.

## 4.3 HIERARCHICAL AND COARSE-TO-FINE PROCESSING

The core principle of processing information at multiple levels of abstraction has deep roots in AI, with a long history of coarse-to-fine and hierarchical methods in computer vision (Fu et al., 2017; He et al., 2017; Feichtenhofer et al., 2019; Xu et al., 2024; Cai & Vasconcelos, 2018). This principle of respecting innate structure is also a cornerstone of Natural Language Processing (Yang et al., 2016; Sarthi et al., 2024; Socher et al., 2013). Selective Deepening formalizes and generalizes this cross-domain pattern into a principle for dynamic *reasoning*. While prior methods often employ fixed hierarchical architectures, Selective Deepening implements this process adaptively at inference time. It leverages a model's intelligence to navigate a structural map and decide where to "zoom in," creating a flexible, modality-agnostic framework.

## 5 CONCLUSION

This work introduced Selective Deepening, a navigational framework for model reasoning that respects and exploits the inherent structure of information. By replacing structure-agnostic retrieval with a two-stage process of abstracting and then deepening, our approach mitigates the context-degradation issues that plague standard methods. Our experiments on function calling and code comprehension demonstrate that this principle yields substantial gains in both reasoning accuracy and computational efficiency.

The principle of Selective Deepening opens several exciting avenues for future research. A primary direction is the development of methods for models to **automatically learn optimal, multilevel abstractions** from raw data, removing the need for predefined schemas. Furthermore, the core navigational capabilities could be enhanced by exploring the use of **multiple, complementary abstractions** simultaneously—akin to using both a table of contents and a semantic index to search a book. Finally, the formal study of **multi-stage recursive deepening** for deeply hierarchical data promises to unlock even more complex, nuanced reasoning. By building on these directions, we can create AI systems that navigate the world's information with ever-greater precision and efficiency.

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
