# OpenReview forum: "Structured Data Understanding: Not All Tokens Are What You Need"
_ICLR.cc/2026/Conference — ICLR 2026 Conference Withdrawn Submission_

### Official Review · Reviewer_qx3u · 2025-10-26

**Soundness:** 2
**Presentation:** 2
**Contribution:** 1
**Rating:** 2
**Confidence:** 5

**Summary:**

Due to the “lost-in-the-middle” and “context rot”, LLMs often struggle to process large amounts of structured information. The authors show that they can improve performance by decomposing the processing into two rounds: (1) first the model looks at the high level structure of the information and selects which sub sections to focus on and (2) the model focuses on those selected regions.

**Strengths:**

- Processing large amounts of structured information is a very important problem.
- The authors evaluate on interesting real-world use cases, including SWE-bench lite and function calling.
- The method is simple and easy to implement with black-box access to LLMs.

**Weaknesses:**

- My main concern is that the work does not propose fundamentally new methodology or surprising results. Nearly all industry coding agents (Claude Code, Codex, OpenHands) process long contexts through a sequence of hierarchical exploration steps (*e.g.* inteleaving `ls` and `grep`). That this leads to improved performance is not surprising and well-documented, for example in the leaderboards for SWE-bench.
- I also cannot evaluate how much Selective Deepening over the state of the art because the work is missing comparisons with nearly all relevant baselines. The only baseline the authors include is the brute-force approach. Here is a non-exhaustive list of baselines that could also be included:
    - RAG
    - Simple agent/tool-based approaches (*e.g. [mini-swe-agent](https://github.com/SWE-agent/mini-swe-agent)*)
    - Divide-and-conquer approaches (*e.g.* [Minions](https://arxiv.org/abs/2502.15964), [LLMxMapReduce](https://arxiv.org/html/2410.09342v1), [Chain of agents](https://scholar.google.com/scholar_url?url=https://proceedings.neurips.cc/paper_files/paper/2024/hash/ee71a4b14ec26710b39ee6be113d7750-Abstract-Conference.html&hl=en&sa=T&oi=gsr-r&ct=res&cd=0&d=15703702160570166190&ei=V0T-aMeaJ5jO6rQPk_O3wAE&scisig=ABGrvjJmpp5NKMY1AArOg0F5MbdL))
    - Memory-based approaches ([MemGPT](https://scholar.google.com/scholar_url?url=https://par.nsf.gov/servlets/purl/10524107&hl=en&sa=T&oi=gsr-r-ggp&ct=res&cd=0&d=8891562156657302957&ei=iET-aPGJM97M6rQPpPXkqAI&scisig=ABGrvjKpjS6UZMfByEhoXTKszw4-), [PRISM](https://scholar.google.com/scholar_url?url=https://arxiv.org/abs/2412.18914&hl=en&sa=T&oi=gsr-r&ct=res&cd=0&d=722654336697936388&ei=bkT-aMngDreO6rQP6NHlsQ0&scisig=ABGrvjK3YvfOohNCybcj2jIxYINt), [Writing-in-the-margins](https://arxiv.org/abs/2408.14906))

**Questions:**

- The title is not clear. I see that the authors are trying to draw a parallel to the “Attention is all you need” paper, but I don’t think that the title makes grammatical sense. Would a more appropriate title be “You do not need all tokens”, or something?
- How would the techniques described in this work generalize to non-coding domains, e.g. a book?

---

### Official Review · Reviewer_UC5f · 2025-11-01

**Soundness:** 4
**Presentation:** 3
**Contribution:** 4
**Rating:** 8
**Confidence:** 4

**Summary:**

This paper introduces “Selective Deepening”, a navigational framework for LLMs long context reasoning including two stages: structural abstraction and navigation and focused reasoning. Evaluation shows this structure-aware navigation framework is broadly applicable and it outperforms baselines on two diverse tasks: function calling and code comprehension, demonstrating that this approach simultaneously improves task accuracy and computational efficiency.

**Strengths:**

1. Importance of problem. The long context code comprehension and processing is a significant problem, especially the “lost in the middle” problem.
2. Clear problem formulation and novel insight. The “selective deepening” idea  is intuitive as it is consistent with how humans process information. Formulating the problem by “navigation over retrieval” is elegant.
3. Solid evaluation. The two evaluation tasks, function calling and code comprehension are appropriate tasks that can reflect the effectiveness of the proposed approach, since both rely on reasoning over deeply structured data.

**Weaknesses:**

1. Sensitivity analysis of navigation is missing. The paper does not explore the sensitivity of the final performance to the Stage 1 selector’s recall, e.g., what if the navigator makes a mistake?
2. Bottleneck of navigators not explored. Recall performance comparison of different models/techniques for navigation is not evaluated.
3. Confusing out-of-scope envisioned techniques. The “recursive deepening” in section 2.1 and “hybrid approach” in section 2.4 looks like some future work ideas that have not been implemented and experimented.
4. Tool and data not publicly available.

**Questions:**

1. How dependent is the approach to the navigation phase? Would the approach remain robust if navigation goes into a wrong direction?
2. The paper mentions that they directly use the powerful reasoning capabilities of foundation models to perform the navigation task, and an alternative approach is to fine-tune a specialized model to maximize the recall. Why the comparison between different navigation techniques is not included in evaluation?
3. Why the “recursive deepening” and “hybrid approach” are in the framework section (section 2)? If they are not implemented, the authors should be upfront, and either move them to the discussion section or highlight what’s the actual implementation and design choice.
4. minor: line 183 “its performance may be challenging” reads strange.

---

### Official Review · Reviewer_Mysq · 2025-11-03

**Soundness:** 1
**Presentation:** 1
**Contribution:** 2
**Rating:** 2
**Confidence:** 4

**Summary:**

Rather than using long-context prompts or RAG to shape the model’s context, this work explores using a “selective deepening” process, where the model takes the overall pool of data and narrows in on relevant aspects. The process is implemented using two passes over the context, one to perform a quick scan and a second to reason over most relevant sub-sections. There are evaluations on SWE-Bench style settings and MCP tool calling settings.

**Strengths:**

Context shaping beyond naive long context and RAG is important towards improving quality. The idea of hierarchically analyzing large corpora and the analogies to human processing are interesting.

**Weaknesses:**

Lack of methodological contributions and descriptions: This paper only articulates a framework, but nothing concrete about how to instantiate it.
- It is not clear how the proposed embedding-based retrieval method works and this is the crux of the paper. L145-L147 says that it leaves the multi-stage abstraction as a direction for future work, when it should be part of this paper / it is what this paper attempts to propose. It’s not clear how the model decides to perform a recursive step – is there a prompt or some other mechanism that leads the reasoning model to decide this.
- It is not clear the proposed method creates the structural abstraction for arbitrary text. Especially because the explanation on L260-266 suggests that the “abstractions retaining sufficient information for selection” is critical for high accuracy, it is important for the paper to describe how abstractions are created.

Experimental protocol: There should be comparisons to neural retrievers for section 3.2.
- It is not clear why the baseline collapse is attributed to the lost in the middle phenomenon (L335-339). Are the missed tools located in the model of the context? That’s not shown. Just because the performance - degrades as there are more tools doesn’t make it “lost in the middle”.
- There is a lack of explanation as to why selective deepening improves over the oracle baseline.
- The choice of context the model chooses using selective deepening versus standard retrieval is not explained via error analysis, so it is unclear where any quantitative performance improvements come from.

Overall, it's not clear what the methodological contribution is and the error analysis for the experiments is insufficient to know why any numbers are the way there are.

**Questions:**

See above.

---

### Official Review · Reviewer_THxo · 2025-11-09

**Soundness:** 3
**Presentation:** 2
**Contribution:** 2
**Rating:** 2
**Confidence:** 4

**Summary:**

The paper proposes Selective Deepening - a methodology by which the language model’s original, flat context is turned into a hierarchical representation, where higher levels are coarse-grained representations of the lower-levels. At each level, the model selects the most relevant areas to “deepen into”, upon which the lower, higher-fidelity level is made available. This mechanism shortens the effective context length while preserving relevant structure, mitigating the “lost in the middle” effect and improves both efficiency and accuracy in long-context reasoning.

For example, in the context of repository-level bug fixing, the model may first make available the class and function signatures of the files, with the implementations hidden. From these, the model picks a subset to “selectively deepen” into, upon which the function bodies of the selected functions are made available for a final bug fix.

The authors test their approach on function calling (BFCLv3) and repository-level bug fixing (SWE-Bench Lite), showing both improved accuracy and efficiency.

**Strengths:**

- The paper **addresses a significant and practically impactful problem**: How to reduce the inefficiency and reasoning degradation that arise when models process long contexts.

- The **main method of the paper is clearly introduced and easy to follow**, with authors providing both illustrative schematics (Figure 1) and rigorous mathematical formalization (section 2.1). I particularly enjoyed the references to sister fields like cognitive science and computer vision in both the introduction and conclusion, helping contextualize the proposed method.

- While Selective Deepening itself, as well as its application to repository-level bug fixing, is not novel, **its use in the context of function calling with large, noisy tool sets is new and well-motivated.**

- The paper **makes a convincing case that Selective Deepening helps mitigate the “lost in the middle” effect.** The experimental design in section 3.2 is well thought-out and clearly isolates the impact of selective deepening under controlled context-length expansions (Figure 4).

**Weaknesses:**

- **The idea of selective deepening is not novel, having already been proposed and validated in the context of repository-level bugfixing in the popular Agentless harness** (Xia et. al., 2024). In addition to the general methodology, section 3.2 of the present work focuses on the same task and benchmark (SWE-Bench Lite) as Agentless while failing to cite it.

- The experiments in the paper focus exclusively on end to end success rates or efficiency improvements, **lacking key quantitative analyses like the navigator’s recall of selected nodes.** The paper would also be strengthened by ablating across variants of navigators - e.g. using a smaller model at specific levels of the hierarchy.

- The paper is **extremely sparse on illustrative examples and reproducibility details** -  clarity would be enhanced by the inclusion of an appendix providing the prompt templates and hyperparameters of each experiment, along with examples of the coarse-grained input at each level of the hierarchy.

- In section 4.2., the authors draw a connection between Selective Deepening and more open-ended LLM agents. **The paper could be strengthened by comparing the proposed approach with fully agentic systems** - this is all the more important as the field has indeed moved from the latter to the former, with the early popularity of the fixed Agentless harness by now largely superseded by fully agentic approaches in the coding domain (c.f. o1 system card, where Agentless is used, with recent model launches).

- In some places, **the paper lacks sufficient level of detail in presenting results**:
  - In section 3.1, the authors mention that they measure efficiency “by the average number of tokens processed per interaction”. However, no quantitative efficiency gain metrics are subsequently provided that’d correspond to the function calling success rates in Figure 2.

  - In Figure 3, the x-axis lists the API cost to illustrate performance improvements. While certainly indicative, the analysis should be extended by a proper accounting of tokens.

  - In section 3.2, it’s mentioned that “experiments were conducted using the DeepSeek-V3 API, which has a 64K context length”. But immediately after,  SWE-Bench Lite with a 100K context length condition was introduced, and results reported on it with Selective Deepening, but with no baseline. Explicitly mentioning the lack of baseline for the 100K condition due to API context length limits would help avoid confusion as to the lack of baseline.

- In section 3.2, **the authors use SWE-Bench Lite as their benchmark, which has been largely abandoned by the community in favor of SWE-Bench Verified** after the former having been reported to have quality issues (e.g. underspecified issue descriptions, overly specific unit tests) leading to many false negatives.

**Questions:**

See "Weaknesses" section

---

### Note · Authors · 2025-12-03

I have read and agree with the venue's withdrawal policy on behalf of myself and my co-authors.